# Particle number concentrations and size distribution in a polluted megacity: The Delhi Aerosol Supersite study

Shahzad Gani[1], Sahil Bhandari[2], Kanan Patel[2], Sarah Seraj[1], Prashant Soni[3], Zainab Arub[3], Gazala Habib[3], Lea Hildebrandt Ruiz[2], and Joshua S. Apte[1]

[1]Department of Civil, Architectural and Environmental Engineering, The University of Texas at Austin, Texas, USA
[2]McKetta Department of Chemical Engineering, The University of Texas at Austin, Texas, USA
[3]Department of Civil Engineering, Indian Institute of Technology Delhi, New Delhi, India

**Correspondence:** Joshua S. Apte (jsapte@utexas.edu), Lea Hildebrandt Ruiz (lhr@che.utexas.edu)

**Abstract.**

The Indian national capital, Delhi, routinely experiences some of the world's highest urban particulate matter concentrations. While fine particulate matter (PM$_{2.5}$) mass concentrations in Delhi are at least an order of magnitude higher than in many western cities, the particle number (PN) concentrations are not similarly elevated. Here we report on 1.25 years of highly time resolved particle size distributions (PSD) data in the size range of 12–560 nm. We observed that the large number of accumulation mode particles—that constitute most of the PM$_{2.5}$ mass—also contributed substantially to the PN concentrations. The ultrafine particles (UFP, D$_p$ <100 nm) fraction of PN was higher during the traffic rush hours and for daytimes of warmer seasons—consistent with traffic and nucleation events being major sources of urban UFP. UFP concentrations were found to be relatively lower during periods with some of the highest mass concentrations. Calculations based on measured PSD and coagulation theory suggest UFP concentrations are suppressed by a rapid coagulation sink during polluted periods when large concentrations of particles in the accumulation mode result in high surface area concentrations. A smaller accumulation mode for warmer months results in increased UFP fraction, likely owing to a comparatively smaller coagulation sink. We also see evidence suggestive of nucleation which may also contribute to the increased UFP proportion during the warmer seasons. Even though coagulation does not affect mass concentrations, it can significantly govern PN levels with important health and policy implications. Implications of a strong accumulation mode coagulation sink for future air quality control efforts in Delhi are that a reduction in mass concentration, especially in winter, may not produce proportional reduction in PN concentrations. Strategies that only target accumulation mode particles (which constitute much of the fine PM$_{2.5}$ mass) may even lead to an increase in the UFP concentrations as the coagulation sink decreases.

## 1 Introduction

Outdoor air pollution has detrimental health effects (Pope and Dockery, 2006; Schraufnagel et al., 2019a) and is responsible for more than 4 million deaths every year globally (Cohen et al., 2017), resulting in substantial global and regional decrements in life expectancy (Apte et al., 2018). There has been a continued growth of megacities (population > 10 million) and as of 2018 there are 47 megacities (United Nations, 2018). Many of the lower-income megacities experience persistently severe air

pollution problems and Delhi (population = 28 million) routinely experiences the highest annual-average fine particle mass (PM$_{2.5}$) concentrations of any megacity in the world (World Health Organization, 2018). In addition to aerosol mass, there is now growing evidence of potential health risks associated with ultrafine particles (UFP, D$_p$<100 nm). These particles can penetrate and deposit deep within the lungs, may enter the bloodstream, and reach sensitive internal organs (Yeh and Schum, 1980; Oberdörster et al., 2005; Salma et al., 2015; Schraufnagel et al., 2019b). While UFPs scarcely contribute to aerosol mass (PM), they have been observed to dominate particle number (PN) concentrations observed in urban environments (Hussein et al., 2004; Rodríguez et al., 2007; Wu et al., 2008).

Atmospheric particles size distributions (PSD) are often categorized by the nucleation mode (3–25 nm), the Aitken mode (25–100 nm) and the accumulation mode (100–1000 nm). UFP include both the nucleation and the Aitken modes. Most of the PM$_{2.5}$ mass is usually contributed from the accumulation mode particles, whereas UFP constitute most of the PN (Seinfeld and Pandis, 2006). Given that different modes of the PSD contribute disproportionately to aerosol mass (accumulation mode for PM$_{2.5}$) and aerosol number (UFP for PN) concentrations, PN concentrations do not necessarily share the same spatial or temporal dynamics as PM (Johansson et al., 2007; Puustinen et al., 2007; Reche et al., 2011). The accumulation mode particles in urban environments result from a wide range of anthropogenic sources (traffic, industrial, biomass burning, etc.), natural sources (wildfires, volcanoes, deserts, etc.), and chemical processing of gases and aerosol (Robinson et al., 2007; Zhang et al., 2007; Jimenez et al., 2009; Calvo et al., 2013). However, traffic (Zhu et al., 2002; Charron and Harrison, 2003; Fruin et al., 2008; Wehner et al., 2009), cooking (Saha et al., 2019; Abernethy et al., 2013), and new particle formation (Brines et al., 2015; Hofman et al., 2016) are generally considered the only major sources of urban UFP.

In cities with high aerosol mass loadings (e.g., Beijing and Delhi), coagulation scavenging can be a major sink of UFP—the numerous but smaller UFP are 'lost' to the fewer but larger particles that contribute most to the aerosol mass (Kerminen et al., 2001; Zhang and Wexler, 2002; Kulmala, 2003; Kulmala and Kerminen, 2008). Specifically, coagulation plays a crucial role in rapid removal of freshly nucleated particles onto pre-existing larger particles (Kerminen et al., 2001; Kulmala, 2003). Short-term studies from Delhi have observed coagulation as a major sink for UFP (Mönkkönen et al., 2004a, 2005), similar to observations from polluted cities in the USA in the 1970s (Husar et al., 1972; Whitby et al., 1972; Willeke and Whitby, 1975) and contemporary polluted cities in China (Cai and Jiang, 2017; Peng et al., 2014; Wu et al., 2008). Polluted megacities such as Delhi often experience aerosol mass concentrations that are an order of magnitude higher than those experienced in cities in high-income countries (Pant et al., 2015; Jaiprakash et al., 2017; Gani et al., 2019), yet PN concentrations are not high in similar proportions (Mönkkönen et al., 2004b; Apte et al., 2011).

As a part of the Delhi Aerosol Supersite (DAS) study we investigated Delhi's aerosol chemical composition and its sources (Gani et al., 2019; Bhandari et al., 2020). Here we use long-term observations from Delhi to present seasonal and diurnal profiles of PSD in this polluted megacity. We also provide insights into the phenomena—such as coagulation—that influence PSD in Delhi and interpret their implications for future policy measures. This study has the potential to help understand processes that drive PSD in other cities with similar sources and meteorology across the Indo-Gangetic plain (Kumar et al., 2017; Singh et al., 2015) and is relevant for other highly polluted urban environments.

## 2 Methods

### 2.1 Sampling site

We installed a suite of high time resolution online aerosol measurement instrumentation at the Indian Institute of Technology Delhi (IIT Delhi) campus in South Delhi. The instruments were located on the top floor of a 4-story building and the nearest source of local emissions is an arterial road located 150 m away from the building. Delhi experiences a wide range of meteorological conditions with large diurnal and seasonal variations in temperature, RH and wind speed, among other parameters (Fig. 1). Furthermore, shallow inversion layers also occur frequently, especially for cooler periods. The prevailing wind direction in Delhi is from the northwest, implying that the air we sampled would have traversed through ~20 km of industrial, commercial, and residential areas in Delhi before reaching our site. We discussed the meteorology of Delhi in further detail in an earlier publication (Gani et al., 2019).

### 2.2 Instrumentation and setup

We measured PSD using a scanning mobility particle sizer (SMPS, TSI, Shoreview, MN) consisting of an electrostatic classifier (TSI model 3080), differential mobility analyzer (DMA, TSI model 3081), X-ray aerosol neutralizer (TSI model 3088), and a water-based condensation particle counter (CPC, TSI model 3785). Chemical composition of non-refractory $PM_1$ (NR-$PM_1$) was obtained using an Aerodyne Aerosol Chemical Speciation Monitor (ACSM, Aerodyne Research, Billerica MA) and BC was measured using a multi-channel aethalometer (Magee Scientific Model AE33, Berkeley, CA). The inlet had a $PM_{2.5}$ cyclone, followed by a water trap and a Nafion membrane diffusion dryer (Magee Scientific sample stream dryer, Berkeley, CA). Further details of the instrumentation, sampling setup, and the data processing methodology for the instrumentation (SMPS, ACSM and aethalometer) and meteorological parameters are provided in Gani et al. (2019).

### 2.3 Data processing and analysis

The SMPS at our site scanned from 12 to 560 nm with each subsequent scan 135 seconds apart. We used the Aerosol Instrument Manager version 9.0 (TSI, Shoreview, MN) software for logging data from the SMPS (software sampling loss corrections turned off). We calculated the transmission efficiency for the DAS sampling line based on the diffusion (Ingham, 1975) and settling (Pich, 1972) losses. We correct the observed PSD for these transmission efficiency losses and calculate the number concentrations (individual size modes and total) and the median diameter based on the updated PSD. Multi-modal lognormal fitting has been widely used to parameterize the PSD (Whitby, 1978; Seinfeld and Pandis, 2006). The parameterization allows for a quantitative description of the PSD and easier comparisons between different PSD data sets. Separating the various PSD modes can also help understand role of sources in driving each mode. We use a multi-modal fitting technique described in Hussein et al. (2005) to estimate individual log-normal modes of the PSD (Sec. 3.6). We included PSD data from January-2017 to April-2018 in this study.

Studies from Asia have estimated aerosol densities in the range 1.3–1.6 g cm$^3$ (Sarangi et al., 2016; Hu et al., 2012). A previous study from Delhi used a density of 1.7 g cm$^{-3}$ to estimate aerosol mass from their PSD observations (Laakso et al., 2006). Based on aerosol composition data, we estimated a density of ~1.5 g cm$^{-3}$ for the non-refractory portion of the aerosol. However, that did not include the BC and metal component of the aerosol mass (we did not measure metals). We assume an overall average particle density of 1.6 g cm$^{-3}$ as it provides a better mass closure (Gani et al., 2019) and is within the uncertainty bounds based on previous density estimates as well. We use this assumed density to estimate the mass in the observed PSD (PM$_{0.56}$), ultrafine particles (PM$_{UFP}$) and accumulation mode particles (PM$_{acc}$).

We calculated the Fuchs form of the Brownian coagulation coefficient ($K_{12}$) using equations published by Seinfeld and Pandis (2006) and then estimated the characteristic coagulation timescale ($\tau_{coag,i}$) of a particle of any size onto particles of any size greater than that particle using the technique of Westerdahl et al. (2009). We calculate condensation sink (CS) for vapor condensing on the aerosol distribution following Kulmala et al. (2001, 1998) and the growth rate (GR) following Kulmala et al. (2012).

$$\tau_{coag,i} = \frac{1}{\sum\limits_{i<j} K_{i,j} N_j} \tag{1}$$

$$CS = 4\pi D \sum_i \beta r_{p,i} N_i \tag{2}$$

$$GR = \frac{\Delta D_p}{\Delta t} = \frac{D_{p2} - D_{p1}}{t_2 - t_1} \tag{3}$$

where $K_{i,j}$ is the coagulation coefficient for particle in bin $i$ of the PSD coagulating with those in bin $j$ and $N_j$ is the concentration of particles in bin $j$ of the PSD. D is diffusion coefficient of the condensing vapor (we use H$_2$SO$_4$ for our calculations), $\beta$ is the transitional correction factor, $r_{p,i}$ is the radius of a particle in bin $i$ of the PSD, and $N_i$ is the concentration of particles in bin $i$ of the PSD. $D_{p1}$ and $D_{p2}$ are the diameters of the particles at times $t_1$ and $t_2$, respectively.

For our analysis, we categorize the seasons as winters (December to mid-February), spring (mid-February to March), summers (April to June), monsoon (July to mid-September) and Autumn (mid-September to November) (Indian National Science Academy, 2018). We define day as 7 AM–7 PM and night as 7 PM–7 AM.

## 2.4 Limitations and uncertainties

Like most measurement studies, the limitations and the uncertainties of this study arise from instrumentation and the sampling system. While we have applied correction for diffusion and settling losses in our sampling system, future studies can reduce sampling losses and subsequent uncertainties—especially for ultrafine particles—by designing shorter sampling systems with higher flow rates. The SMPS used for the PSD measurements has a ~30% uncertainty for submicron aerosol (Buonanno et al., 2009). Furthermore, we only measure particles larger than 12 nm, potentially underestimating concentrations of nucleation mode particles. Our study does not provide any information on nanocluster aerosol (1–3 nm) which can contribute

to a significant fraction of PN concentrations and their detection made possible by recent advances in aerosol measurement instrumentation (Rönkkö et al., 2017; Kontkanen et al., 2017; Kangasluoma and Attoui, 2019). Measuring the complete nucleation mode range along with these nascent nanoclusters can advance our knowledge of particle formation pathways and their subsequent growth that leads to high air pollution (Yao et al., 2018; Guo et al., 2014).

Another limitation of this study is that our measurements were limited to a single site, and therefore do not capture the spatial variability within Delhi. UFP concentrations can be quite variable even within a small spatial domain (Saha et al., 2019; Puustinen et al., 2007), so it is likely that a megacity like Delhi with diverse local sources will have strong spatially variability in PSD and PN concentrations. Future studies can quantify this spatial variability by measuring PSD at multiple fixed sites and using techniques such as mobile monitoring (e.g., Apte et al. (2017)).

## 3   Results and discussion

### 3.1   Particle number and mass concentrations

Delhi experiences large seasonal and diurnal variations in aerosol mass concentrations. In Gani et al. (2019), we showed that the wintertime submicron aerosol concentration was ~2× higher than spring and ~4× higher than the warmer months. In Fig. 2 we compare the seasonal changes in aerosol mass and PN concentrations. In contrast to the sharp seasonal variation in aerosol mass loadings, PN levels had much lower variability. The average PN levels for winter were 52500 cm$^{-3}$, spring 49000 cm$^{-3}$, summer 43400 cm$^{-3}$, monsoon 35400 cm$^{-3}$, and autumn 38000 cm$^{-3}$. The differences in the variability of number and mass concentrations are potentially explained by the difference in the sources and processes that drive their concentrations (Sec. 1).

For each season, PN concentrations varied by the time of day (Fig. 3). In winter, average hourly PN concentrations ranged between 32400 cm$^{-3}$ and 77500 cm$^{-3}$. The hourly averaged winter PN levels had a morning (08:00) peak, followed by lowest concentrations during the afternoon (14:00), and highest concentrations during the evening (21:00). The spring hourly averaged concentrations ranged between 38100 cm$^{-3}$ and 62400 cm$^{-3}$. The hourly averaged spring levels also had the morning (08:00) and evening (21:00) peaks, and the PN concentrations were lowest around 15:00. Summer hourly averaged concentrations ranged between 35900 cm$^{-3}$ and 54200 cm$^{-3}$. The hourly averaged summer levels also had the morning (08:00) and evening (21:00) peaks, there was also a small peak during the midday (11:00) and the PN concentrations were lowest around 03:00. Monsoon hourly PN concentrations ranged between 25800 cm$^{-3}$ and 43500 cm$^{-3}$. The monsoon PN hourly average concentrations were lowest during early morning (04:00), highest during the midday (12:00), followed by another evening peak in the evening (21:00). Autumn hourly average PN concentrations ranged between 25500 cm$^{-3}$ and 49200 cm$^{-3}$. The autumn diurnal profile had a morning (07:00) and an evening peak (21:00), with the concentrations lowest during the daytime (13:00). While hourly averaged PN concentrations for all seasons had peaks during the morning and the evening, additional midday peaks were observed during the summer and monsoon.

In Fig. 3 we also compare the seasonal and diurnal changes in PN and PM$_{0.56}$ concentrations—number and mass concentrations observed from the SMPS. The SMPS-based mass concentrations follow the diurnal profile similar to those of aerosol mass loadings in Delhi (Gani et al., 2019). However, PN diurnal profiles do not closely follow those of PM$_{0.56}$ (campaign R$^2$

of the hourly averaged time-series of PN and $PM_{0.56} = 0.26$). The low correlation of aerosol number and mass is consistent with UFP making up a large fraction of the PN (Sec. 3.2), but contributing much less to the aerosol mass compared to larger accumulation mode particles. In Fig. 4, we estimate the aerosol mass loading in the ultrafine mode and the accumulation mode particles. While $PM_{acc}$ generally accounted for more than 90% of the $PM_{0.56}$ concentrations, the $PM_{UFP}$ fraction of $PM_{0.56}$ ranged between 3 and 10% depending on season and time of day. The fractional $PM_{UFP}$ contribution to $PM_{0.56}$ was highest during the evening traffic rush hours for the summer and monsoon, and lowest during the non-traffic rush hours of the winter and autumn. The average wintertime $PM_{UFP}$ concentrations ranged from 2.8 µg m$^{-3}$ for midday to 10.3 µg m$^{-3}$ during the evening traffic. For the other seasons, average $PM_{UFP}$ concentrations were in the range ~2–5 µg m$^{-3}$. Overall, the mass contributions of UFP ranged between 2.1 and 10.3 µg m$^{-3}$ depending on season and time of day. For contrast, mass concentrations of UFP have been observed to be <3 µg m$^{-3}$ in Germany (Brüggemann et al., 2009), <2 µg m$^{-3}$ in Taiwan (Cheung et al., 2016), and <0.2 µg m$^{-3}$ in California (Xue et al., 2019). Furthermore, even if the fractional contribution to total mass in Delhi remains low (<10%), the absolute mass concentrations from the observed UFP during polluted periods were nearly as high as the $PM_{2.5}$ concentrations observed in many cities in North America (~8–12 µg m$^{-3}$) (Manning et al., 2018).

### 3.2   Size resolved particle concentrations

The PN concentrations and mass loadings are driven by the concentrations of the individual size modes—nucleation ($N_{nuc}$), Aitken ($N_{ait}$), and accumulation ($N_{acc}$). It should be noted that the $N_{nuc}$, $N_{ait}$, and $N_{acc}$ refer to the specific particle size range and not the individual lognormal modes that constitute the PSD (Sec. 3.6). While the $N_{nuc}$ and $N_{ait}$ constitute the UFP concentrations (UFP = $N_{nuc}$ + $N_{ait}$), $N_{acc}$ contributes to both the total PN concentrations (PN = $N_{nuc}$ + $N_{ait}$ + $N_{acc}$) and most of the fine aerosol mass (Seinfeld and Pandis, 2006). In Fig. 3 we present the seasonal and diurnal variation of $N_{nuc}$, $N_{ait}$, $N_{acc}$, UFP, PN, and the median particle diameter. In Fig. 5 we present the stacked absolute and fractional diurnal profiles of the PN components for each season to illustrate the varying contribution of each size mode to the PN concentrations by season and time-of-day. Furthermore, in Table 1 we present the summary daytime and nighttime averages for all seasons.

The nucleation mode particle concentrations were highest (in magnitude and fraction of PN) during the warmer periods, when particle mass concentrations tend to be lower. The average hourly $N_{nuc}$ peaked around 11:00–12:00 for spring, summer, and monsoon. These $N_{nuc}$ peak hourly averaged concentrations were ~18000 cm$^{-3}$ and contributed to almost 40–50% of the PN concentrations for these periods. For nighttime of spring, summer and monsoon and for all times of day for autumn and spring, hourly averaged $N_{nuc}$ concentrations were usually less than 10000 cm$^{-3}$ and contributed to within 10–20% of the PN concentrations. The low particle mass concentrations and high insolation during the daytime of warmer months act together to favor nucleation and survival of nucleated particles (Kerminen et al., 2018), resulting in higher $N_{nuc}$ concentrations during these periods. In Sec. 3.3, we discuss the strong condensation and coagulation sink during extremely polluted periods (such as winter and autumn) and its role in suppressing $N_{nuc}$ concentrations.

The Aitken mode particle concentrations generally constituted the largest fraction of the PN concentrations. $N_{ait}$ contributed to 40–60% of PN concentrations depending on season and time of day. $N_{ait}$ concentrations were highest during the morning and evening hours—periods with lower ventilation and high vehicular traffic. The average hourly $N_{ait}$ concentrations ranged

between 12000 cm$^{-3}$ (monsoon early morning) to 44300 cm$^{-3}$ (winter evening). The hourly averaged $N_{ait}$ profiles for most seasons had a morning and an evening peak. For winter, autumn, and spring, the hourly averaged $N_{ait}$ concentrations were lowest during the midday. However, for summer and monsoon there was another peak during the midday with lowest levels observed early in the morning. The small daytime $N_{ait}$ peaks for the warmer months are consistent with nucleation mode

particles growing to the Aitken mode (Kulmala et al., 2004). The $N_{ait}$ peaks during the morning and the evening traffic rush hours are consistent with vehicular emissions contributing in this size range (Paasonen et al., 2016; Kumar et al., 2010). Studies from other cities—especially from near-roadway sites—also observe large fraction of the Aitken mode particles in the total PN concentrations (Zhu et al., 2002; Wu et al., 2008). In addition to traffic, other sources contributing to Aitken mode particles include cooking, industrial activities, solid waste burning, and construction activities (Kumar et al., 2013; Riffault et al., 2015;

Chen et al., 2017).

Accumulation mode particles contribute to almost all of the submicron aerosol mass, so it is unsurprising that the diurnal and seasonal variation of $N_{acc}$ concentrations are similar to those of aerosol mass concentrations (Gani et al., 2019). In addition to contributing to most of the fine PM mass concentrations, $N_{acc}$ in Delhi also constituted a significant fraction of the PN concentrations. The $N_{acc}$ fraction of PN were the highest for nighttimes of autumn and winter (~45%) and the lowest for the

daytimes of the warmer summer and monsoon seasons (~15%). The average hourly $N_{acc}$ concentrations ranged between 4200 cm$^{-3}$ (monsoon midday) to 26200 cm$^{-3}$ (winter evening). For all seasons, the hourly averaged hourly $N_{acc}$ concentrations had a morning (07:00) peak and an evening (20:00–22:00) peak, with lowest concentrations during the daytime (13:00–15:00). In addition to emitting in the Aitken mode, biomass burning and older diesel vehicles emit in the accumulation mode as well (Chen et al., 2017; Paasonen et al., 2016; Kumar et al., 2010). Additionally, particles from fresh vehicular emissions can grow

rapidly from the Aitken mode at the tailpipe to accumulation mode at roadside and ambient locations (Ning et al., 2013). We speculate that the traffic-related particles in Delhi may get smaller as India's vehicle fleet (especially heavy-duty trucks) is upgraded (Guttikunda and Mohan, 2014; Zhou et al., 2020a).

Overall, UFP contributed to ~65% of the PN concentrations for winter and autumn. The UFP fraction of PN concentrations was relatively higher for spring (75%), summer (78%), and monsoon (80%). The average hourly UFP concentrations ranged

between 17200 cm$^{-3}$ (autumn midday) to 52500 cm$^{-3}$ (winter evening). For all seasons, UFP concentrations had peaks in the morning (07:00–09:00) and the evening (19:00–21:00). For the relatively warmer seasons (summer, monsoon, and spring), there were additional peaks during the daytime (11:00–13:00). The UFP fraction of PN (UFP/PN) was generally the highest during the daytime in spring, summer, and monsoon (~85%). Conversely, some of the lowest UFP/PN levels were observed during the nighttime of winter and autumn (~55%). The UFP/PN levels observed in Delhi are generally lower than those in

cleaner cities (Hussein et al., 2004; Rodríguez et al., 2007; Putaud et al., 2010) and even lower compared to some other polluted ones where mass concentrations often exceeded ~100 μg m$^{-3}$ (Whitby et al., 1975; Laakso et al., 2006; Wu et al., 2008). As a result of the relatively large fraction of PN being constituted of non-UFP ($N_{acc}$) in Delhi, the median diameter size observed were often much larger than those observed in these cleaner cities.

The hourly averaged median diameters were largest during nighttime of winter and autumn, reaching up to ~90 nm. The

smallest hourly averaged median diameters (~40 nm) were observed during the daytime of spring, summer, and monsoon. As

with the concentrations and fractions of the PN modes, the median diameters also had sharp seasonal and diurnal variations. For winter and autumn, the hourly averaged median diameters ranged between 50 nm during the late afternoon (16:00) to 90 nm during the late night/early morning (03:00). The median diameters were the smallest for the relatively less polluted warmer months (summer, monsoon, and spring), with the hourly averaged median diameters ranging between 35 nm (12:00) to 75 nm (03:00). For comparison, the median diameter in Helsinki (calculated for ~12–560 nm) was ~30 nm (Hussein et al., 2004). In Los Angeles, the median diameters up to 150 m from a freeway (our site is ~150 m from an arterial road) were <50 nm (Zhu et al., 2002). We found that for all seasons, the nighttime had larger particles than the daytime. For winter and autumn, the average median diameters were ~65 nm during the day and ~80 nm during the night. For the spring, summer, and monsoon, the average median diameters were ~40 nm during the day and ~65 nm during the night.

In Fig. 6, we present the average observed PSD evolving over the day for each season as a heatmap. For the spring, summer, and monsoon the average seasonal heatmap indicates daytime new particle formation. However, winter and autumn had lower concentrations of the smaller particles that are generally associated with new particle formation. In the following section (Sec. 3.3) we explore the role of coagulation scavenging during polluted periods in selectively suppressing concentrations of smaller particles and resulting in an increase in the median diameter size.

## 3.3 Coagulation scavenging

The evolution of ambient particle size distributions reflects the complex interplay between emissions, atmospheric dilution, and a wide range of aerosol dynamic processes including new particle formation, evaporation/condensation, and coagulation. In this section we explore the role of coagulation scavenging in suppressing both the existing UFP concentrations and new particle formation during periods with high aerosol mass loadings corresponding to elevated $N_{acc}$ levels. During the polluted winter and autumn seasons, the daytime UFP levels were often lower than those during the same period in the warmer months.

In Fig. 7 we present PSD heatmaps for an extremely polluted (high aerosol mass loading) period (Feb 1–5, 2017). During the polluted episode (Fig. 7 (a)), the $PM_{0.56}$ concentrations ranged between 100 and 400 μg m$^{-3}$ and the PN concentrations between 25000 and 80000 cm$^{-3}$. The peak PN levels were observed during the evening traffic rush hours. Even though PN concentrations decreased during the late evening (~8–11 PM), the $PM_{0.56}$ levels kept continuously increasing all night. The $N_{nuc}$ levels decreased during these polluted evening periods even as $N_{ait}$ and $N_{acc}$ levels increased (Fig. S13). This dynamic suggests that coagulation may be acting as a strong control on concentrations of smaller particles, even as emissions are high. The characteristic timescale for particles smaller than 15 nm to be lost to coagulation scavenging by larger particles was less than 15 minutes for these polluted evenings. The size of the particles rapidly increased from the evening to the nighttime (growth rates up to ~10 nm h$^{-1}$) and coagulation may be one of the mechanisms that contributes to this rapid growth.

Since coagulation only affects particle number concentrations, and not mass concentrations, $PM_{0.56}$ increased into the night, possibly owing to a combination of increased sources (e.g., nocturnal truck traffic, biomass burning for cooking and heat) and decreasing ventilation (Guttikunda and Gurjar, 2012; Guttikunda and Calori, 2013; Bhandari et al., 2020). The PN concentrations dropped after the evening traffic rush hours and only increased again during the morning traffic rush hours.

We followed Westerdahl et al. (2009) to calculate the coagulation timescales corresponding to the observed PSD (Sec. 2.3). When a small particle coagulates onto a larger particle, the small particle is lost and the big particle scarcely grows. In Fig. 8 we have presented the diurnal and seasonal profiles for the modeled coagulation timescales for a 15, 30, and 100 nm particle. For the nucleation mode particle (15 nm), the timescales ranged from a few minutes for the polluted periods (mornings and evenings of cooler seasons) to ~1 h for the relatively less polluted periods (midday of warmer seasons). The corresponding range for an Aitken mode particle (30 nm) was ~1–4 h. Finally, for an accumulation mode particle (100 nm), coagulation was not a significant sink owing to the extremely long timescales (tens of hours). Advection timescales across Delhi are ~3–5 h based on a length scale of ~40 km at typical wind speeds.

Coagulation scavenging of UFP on to the larger accumulation mode particles could explain the suppression in UFP concentrations for polluted periods which have high accumulation mode concentrations. The high aerosol surface area from particles in the accumulation mode, in addition to contributing to most of the mass, can act as a coagulation sink for the smaller particles. The scavenging of UFP in Delhi during the polluted periods could explain the higher median particle diameters than those usually observed in other urban environments (Salma et al., 2011). In Table 2, we compare characteristic condensation sink ($H_2SO_4$) and coagulation sink (for 1, 5, 10, 15, 30, and 100 nm particles) for Delhi (least and most polluted) with other cities (clean and polluted). Based on the coagulation timescales for these polluted periods, for a 1 h period coagulation scavenging could result in removal of ~85% of the 10 nm particles, ~50% of the 30 nm particles, and ~10% of the 100 nm particles present at the beginning of the hour. The coagulation sink for UFP in Delhi during the polluted periods was ~20 times larger as compared to a clean city (Helsinki). Even for least polluted periods in Delhi, the coagulation sink for UFP was ~4 times larger than Helsinki.

## 3.4 New particle formation

For the less polluted seasons (Fig. 6), we observed a sharp increase in the concentration of nucleation mode particles during middays suggesting new particle formation. In Fig. 7 (b) we present PSD heatmaps for a relatively less polluted period (Apr 7–11, 2017). During some days for this period we observed new particle formation and growth. The growth of particles in the nucleation mode was especially prominent for the latter two days of this episode when the $PM_{0.56}$ concentrations were almost half the $PM_{0.56}$ concentrations of the first two days and consequently both the condensation and coagulation sinks (across particle sizes) also decreased by ~50%. We observed "banana-shaped" new particle formation and growth events during these relatively clean conditions which were consistent with growth of atmospheric nanoparticles observed elsewhere (Kulmala and Kerminen, 2008). The growth rate of the nucleation mode particles based on the average PSD for these clean conditions was ~5 nm h$^{-1}$. It should be noted that our SMPS measurements had a lower size cut-off of 12 nm, implying that the particles in the nucleation mode had already grown before being detected.

Growth rates depend on particle size, concentrations of condensable gases ($H_2SO_4$, highly oxygenated molecules, etc.), and condensation and coagulation sinks, and they can range from 0.1 to 10s of nm h$^{-1}$ across various environments (Kerminen et al., 2018; Nieminen et al., 2018; Bianchi et al., 2016). We did not observe "banana-shaped" new particle formation and growth events during the polluted winter or autumn seasons as the growth phase was probably disrupted by coagulation scavenging.

For example, during winter, coagulation timescales for a 15 nm particle were <1 h (Fig. 8). Growth rates for polluted megacities also usually slower than 10 nm h$^{-1}$ (Zhou et al., 2020b; Chu et al., 2019). It is likely that a nucleation mode particle is more likely to coagulate with a larger particle than to grow to the Aitken/accumulation mode. Overall, the aerosol dynamics are a complex interplay of both new particle formation and coagulation scavenging, with nucleation mode particles in highly polluted

environments generally susceptible to scavenging by the accumulation mode particles.

Large accumulation mode concentrations also act as a strong condensation sink (Table 2), causing vapors to condense on to existing particles instead of forming new particles (Mönkkönen et al., 2004a). During the most polluted time in Delhi (generally 8–9 PM during the winter), the condensation sink was 20 times larger than Helsinki (clean city) (Hussein et al., 2004). Even for the least polluted period in the Delhi (generally 2–3 PM during the monsoon), the condensation sink was ~4 times larger

than Helsinki. The condensation sink for the polluted periods was somewhat higher than Beijing (polluted city) (Laakso et al., 2006), but within the same order of magnitude. For spring, summer, and monsoon, we estimated the condensation sink to be more than an order of magnitude lower than the winter and autumn. Consequently, unlike winter and autumn, we observed an increase in daytime UFP concentrations (around 12–30 nm) for the relatively less polluted months which was consistent with daytime new particle formation from nucleation events (Kulmala et al., 2004; Brines et al., 2015). Based on the survival

parameter ($P = (CS/10^4 \text{ s}^{-1})/(GR/\text{nm h}^{-1})$) defined in Kulmala et al. (2017), new particle formation would not be expected even during some of the least polluted periods in Delhi. Why we observe any new particle formation events in Delhi—similar to observations from megacities in China—is still not well understood (Kulmala et al., 2017).

## 3.5  Role of meteorology

In Gani et al. (2019) we showed that meteorology—specifically change in ventilation—was an important factor in driving the

seasonal aerosol mass variations observed in Delhi. However, PN levels did not show similar seasonal variation (Fig. 2). In Fig. S11, we have presented the 2-week averages for ventilation and PN concentrations (by mode and total). $N_{acc}$ concentrations decrease as the ventilation coefficient increases. Since accumulation mode particles constitute most of the submicron mass, decrease in $N_{acc}$ concentrations with increasing ventilation is consistent with our previous analysis that showed decrease in mass loadings with increasing ventilation. $N_{ait}$ concentrations seem to have a smaller slope than accumulation mode concentrations.

$N_{nuc}$ concentrations increased with increasing ventilation, consistent with new particle formation and/or less coagulation scavenging. Comparing rainy and non-rainy days during the monsoon, our observations suggest that rain reduces the accumulation mode concentrations (Fig. S12) as has been observed in other studies (Luan et al., 2019; Hussein et al., 2018). However, we do not observe rain to cause a clear reduction in either nucleation or Aitken mode concentrations. Parametrization studies suggest that rain is more effective in scavenging nucleation mode particles as compared to Aitken mode particles (Pryor et al., 2016;

Laakso et al., 2003). Lower accumulation mode concentrations following rain events imply lower condensation and coagulation sinks. While we cannot fully reconcile our observations here with theory, possible explanations include (i) differential water solubility by particle size and age, or (ii) that the rain washout of the nucleation mode particles is 'balanced' by the reduction in the condensation and coagulation sink due to rain washout of accumulation mode particles.

The lower sensitivity of PN to ventilation changes can be explained by the dissimilar sources and atmospheric process that contribute to PN and PM levels. While PN is mostly comprised of UFP, accumulation mode particles constitute most of the fine aerosol mass. Traffic, cooking, and nucleation events contribute to urban UFP. A large fraction of accumulation mode particles in contrast result from biomass burning and aging of aerosol which result in particles often larger than 100 nm (Janhäll et al., 2010). Periods with lower ventilation that result in high aerosol mass loadings also create a large coagulation sink for UFP (Sec. 3.3). While coagulation suppresses the PN levels, the aerosol mass remains conserved. While a decrease in ventilation should initially increase concentrations for all particle sizes similarly, coagulation causes the smaller UFP to be selectively 'lost'—causing a larger increase for aerosol mass compared to number. Conversely, periods with high ventilation (daytime of warmer months) also have more nucleation events resulting in increased UFP concentrations, but with almost no contribution to aerosol mass (Brines et al., 2015). In addition to large condensation sinks from existing particles (Sec. 3.4), it is possible that the elevated RH during the cooler winter and autumn (Fig. 1) could further suppress atmospheric nucleation. It has been observed in both clean and polluted environments that the RH during new particle formation event days tends to be lower compared to non-event days (Kerminen et al., 2018; Dada et al., 2017; Hamed et al., 2011). Overall, ventilation strongly influences aerosol mass loadings in Delhi, with higher ventilation often resulting in reduction of PM concentrations and vice versa. However, we do not observe such a simple relationship between ventilation and PN levels. The complex interplay of sources, aerosol dynamics, and atmospheric processes as discussed above make PN levels less sensitive to meteorological changes as compared to aerosol mass loadings.

## 3.6 Lessons from multi-modal PSD fitting

In this section we present the parameters obtained by fitting multi-modal lognormal distributions on the seasonal and characteristic time-of-day averages of the observed PSD (Table 3). For most seasons and times of day, the PSD were bimodal. Even when there was a third mode, it contained less than 10% of the total PN concentration. While the smaller mode (ultrafine mode) ranged from 20 to 40 nm depending on season and time of day, the larger mode (accumulation mode) ranged from 80 to 120 nm. For all seasons the late night ultrafine mode concentrations contributed to less than half of the total PN concentration. This observation can be explained by fewer sources of UFP—less traffic and no nucleation events—and a large coagulation sink from nighttime aerosol mass loadings (from accumulation mode) which are often high in part due to unfavorable nighttime meteorology (Gani et al., 2019). Delhi traffic usually peaks around 8–11am in the morning and 5–9pm in the evening. Restrictions on daytime heavy duty truck traffic have resulted in increased nocturnal truck traffic after 10pm (Mishra et al., 2019; Guttikunda and Calori, 2013). The accumulation mode during late nights had median diameters between 100 nm for the relatively less polluted summer and monsoon to 120 nm for spring, winter, and autumn.

During the morning rush hour, the median diameter of the ultrafine and accumulation mode were the smallest for the summer (21, 100 nm for ultrafine and accumulation mode) and monsoon (23, 110 nm) compared to spring (28, 122 nm), autumn (35, 120 nm), and winter (39, 122 nm). Autumn had three modes (35, 120, 234 nm) with the ultrafine mode contributing to 58%, accumulation mode to 34% and the third mode contributing to only 8% of the total PN concentration. For winter during the same period, half of the PN concentrations were in the ultrafine mode (~40 nm) and the other half in the accumulation mode

(~120 nm). The fraction of PN in ultrafine mode during the same period was relatively higher for the other seasons which had less aerosol mass loadings compared to winter, with the least polluted monsoon having 66% of the PN in the ultrafine mode.

The midday PSD were bimodal for winter (26, 125 nm), autumn (29, 122 nm), spring (19, 100 nm), and summer (26, 111 nm). For monsoon the PSD was trimodal (27, 90, 177 nm), with only 6% of the PN fraction in the third mode. Generally, the midday hour had some of the highest fraction of PN concentrations in the ultrafine mode, especially for summer (80%), monsoon (83%), and spring (81%). The corresponding levels while much lower for the more polluted autumn (56%) and winter (53%), were higher than most other times of day within those seasons. Even in absolute terms, summer, monsoon, and spring had some of the highest PN concentrations in the ultrafine modes. The high ultrafine mode concentrations—both magnitude and PN fraction—is potentially from new particle formation (Kulmala et al., 2004). We observe characteristic new particle formation followed by some growth for the warmer seasons (Fig. 6). However, during the more polluted winter and autumn, daytime new particle formation is not observed—potentially because of strong coagulation and condensation sink (Sec 3.4).

During the evening rush hour, the PSD were bimodal for winter (35, 88 nm), autumn (39, 103 nm), spring (28, 80 nm), and summer (30, 89 nm). For monsoon the PSD was trimodal (27, 76, 203 nm), with only 5% of the PN fraction in the third mode. The relatively smaller median diameter for the accumulation mode during the traffic rush hour is consistent with particles from fresh vehicle exhaust—both gasoline and diesel—being smaller than those from biomass burning and other aged aerosol. While gasoline engines emit in the ~20–60 nm range, some of the older diesel engines still prevalent in India can emit larger accumulation mode particles (Kumar et al., 2010). Particles from biomass burning are in the ~50–200 nm size range (Chen et al., 2017; Janhäll et al., 2010). For the warmer periods which have relatively less biomass burning emissions, the ultrafine modes contribute to relatively higher fraction of PN—especially during traffic rush hours. These findings are consistent with studies from cities in high-income countries where traffic and nucleation events are considered the major source of UFP (Brines et al., 2015).

## 4    Conclusions

We used continuous, highly time-resolved and long-term data to provide a detailed seasonal and diurnal characterization of Delhi's PN concentrations. The number concentrations for each mode (nucleation, Aitken, and accumulation) varied dynamically by season and by time of day. While Delhi experiences PM levels 10 times higher than cities in North America and Western Europe, PN levels were comparable to those observed in urban sites in relatively cleaner (in terms of aerosol mass) cities. Observations from other polluted cities (e.g., in China) also show that high aerosol mass does not necessarily imply similarly high PN concentrations (Shen et al., 2011; Wu et al., 2008; Laakso et al., 2006). Furthermore, the seasonal variability of PN was much less than that of the PM measured at our site. While it is generally assumed that UFP constitute most of the PN concentrations, we observed that large number of accumulation particles—that constitute most of the fine aerosol mass—contributed to almost half of the PN concentrations for some of the extremely polluted periods. UFP concentrations were found to be lower during periods with some of the highest mass concentrations.

We show that the lack of proportionality between aerosol mass and number concentrations result from rapid coagulation of UFP, especially during periods with high accumulation mode concentrations. Furthermore, the accumulation mode particles can also act as a strong condensation sink, causing vapors to condense onto existing particles instead of forming new particles. Even though coagulation does not affect mass concentrations, it can significantly govern PN levels with important health and policy implications. Furthermore, implications of a strong accumulation mode coagulation sink for future air quality control efforts in Delhi are that a reduction in mass concentration, may not produce proportional reduction in PN concentrations. Long term continuous observations of PSD from Delhi will be able to provide important insights into the role of sources and atmospheric processes that drive aerosol number concentrations.

*Data availability.* Hourly PSD data used in this study will be made available via the Texas Data Repository upon publication.

*Author contributions.* JSA, LHR, GH, SG and SB designed the study. SG, SB, PS, ZA and SS carried out the data collection. SG carried out the data processing and analyses. All co-authors contributed to interpretation of results, writing, and reviewing the manuscript.

*Competing interests.* The authors declare that they have no conflict of interest.

*Acknowledgements.* JSA was supported by the Climate Works Foundation. We are thankful to the IIT Delhi for institutional support. We are grateful to all student and staff members of the Aerosol Research Characterization laboratory (especially Nisar Ali Baig and Mohammad Yawar Hasan) and the Environmental Engineering laboratory (especially Sanjay Gupta) at IIT Delhi for their constant support. We are thankful to Maynard Havlicek (TSI) for always providing timely technical support for the instrumentation. Tareq Hussein provided the scripts used for mode fitting of the particle size distribution.

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

**Table 1.** Day and night summary of observations derived from the PSD and meteorological parameters. Arithmetic mean used for all species and parameters, except wind direction for which we used median to estimate its central tendency.

| | Winter | | Spring | | Summer | | Monsoon | | Autumn | |
|---|---|---|---|---|---|---|---|---|---|---|
| | Day | Night | Day | Night | Day | Night | Day | Night | Day | Night |
| $N_{nuc}$[a] ($\times 10^3$ cm$^{-3}$) | 7.6 | 5.0 | 15.8 | 9.3 | 13.4 | 7.8 | 13.7 | 6.9 | 6.1 | 3.8 |
| $N_{ait}$[a] ($\times 10^3$ cm$^{-3}$) | 21.3 | 31.4 | 19.7 | 27.1 | 19.6 | 24.2 | 18.0 | 17.6 | 17.1 | 20.9 |
| $N_{acc}$[a] ($\times 10^3$ cm$^{-3}$) | 13.4 | 22.8 | 8.4 | 14.8 | 7.0 | 12.3 | 4.9 | 7.7 | 9.8 | 15.8 |
| UFP[b] ($\times 10^3$ cm$^{-3}$) | 28.9 | 36.4 | 35.5 | 36.4 | 33.0 | 32.0 | 31.7 | 24.5 | 23.2 | 24.7 |
| PN[c] ($\times 10^3$ cm$^{-3}$) | 42.3 | 59.2 | 43.9 | 51.2 | 40.0 | 44.3 | 36.6 | 32.2 | 33.0 | 40.5 |
| UFP/PN | 0.68 | 0.62 | 0.81 | 0.71 | 0.83 | 0.72 | 0.87 | 0.76 | 0.70 | 0.61 |
| Median $D_p$ (nm) | 63.1 | 81.2 | 38.5 | 64.1 | 43.2 | 63.4 | 39.5 | 58.6 | 63.1 | 80.3 |
| SA ($\times 10^3$ μm$^2$cm$^{-3}$) | 2.04 | 3.24 | 1.25 | 2.12 | 0.99 | 1.65 | 0.74 | 1.06 | 1.51 | 2.21 |
| Vol (μm$^3$cm$^{-3}$) | 84.1 | 128 | 47.5 | 81.0 | 34.3 | 58.6 | 25.9 | 37.3 | 62.1 | 87.1 |
| PM$_{0.56}$[d] (μg m$^{-3}$) | 135 | 204 | 76.0 | 130 | 54.9 | 93.7 | 41.4 | 59.8 | 99.4 | 139 |
| | | | | | | | | | | |
| Temperature (°C) | 17 | 13 | 26 | 21 | 35 | 31 | 32 | 29 | 28 | 23 |
| Relative Humidity (%) | 60 | 78 | 42 | 59 | 34 | 43 | 71 | 81 | 49 | 67 |
| Wind Speed (ms$^{-1}$) | 2.7 | 2.6 | 3.2 | 2.6 | 3.8 | 2.9 | 3.4 | 2.5 | 2.5 | 2.1 |
| Wind Direction (°N) | 300 | 300 | 300 | 300 | 270 | 270 | 250 | 190 | 300 | 250 |
| PBLH (m) | 920 | 340 | 1800 | 1000 | 2400 | 1600 | 1600 | 460 | 1460 | 880 |

[a]The modes are based on SMPS observations—nucleation ($12 < D_p < 25$ nm), Aitken ($25 < D_p < 100$ nm) and accumulation ($100 < D_p < 560$ nm) modes. [b]UFP = $N_{nuc}$ + $N_{ait}$. [c]PN = UFP + $N_{acc}$. [d]Estimation of PM$_{0.56}$ concentrations were based on the volume concentrations observed by the SMPS and assuming particle density to be 1.6 g cm$^{-3}$.

**Table 2.** Characteristic coagulation and condensation sinks for Delhi and comparison with those calculated using PSD in literature from other cities.

| | | DAS (our study) | | Literature | | |
|---|---|---|---|---|---|---|
| | | Most polluted[a] | Least polluted[b] | Delhi[c] | Beijing[c] | Helsinki[d] |
| Condensation sink (s$^{-1}$) | H$_2$SO$_4$ | $1.7 \times 10^{-1}$ | $3.5 \times 10^{-2}$ | $1.4 \times 10^{-1}$ | $1.1 \times 10^{-1}$ | $8.2 \times 10^{-3}$ |
| | 1 nm | $8.7 \times 10^{-2}$ | $1.8 \times 10^{-2}$ | $7.3 \times 10^{-2}$ | $5.7 \times 10^{-2}$ | $4.3 \times 10^{-3}$ |
| | 5 nm | $5.7 \times 10^{-3}$ | $1.3 \times 10^{-3}$ | $4.6 \times 10^{-3}$ | $3.4 \times 10^{-3}$ | $3.1 \times 10^{-4}$ |
| Coagulation sink (s$^{-1}$) | 10 nm | $1.8 \times 10^{-3}$ | $4.2 \times 10^{-4}$ | $1.4 \times 10^{-3}$ | $1.0 \times 10^{-3}$ | $1.1 \times 10^{-4}$ |
| | 15 nm | $9.2 \times 10^{-4}$ | $2.3 \times 10^{-4}$ | $7.2 \times 10^{-4}$ | $5.2 \times 10^{-4}$ | $5.6 \times 10^{-5}$ |
| | 30 nm | $3.2 \times 10^{-4}$ | $7.4 \times 10^{-5}$ | $2.4 \times 10^{-4}$ | $1.7 \times 10^{-4}$ | $1.9 \times 10^{-5}$ |
| | 100 nm | $3.7 \times 10^{-5}$ | $6.4 \times 10^{-6}$ | $3.0 \times 10^{-5}$ | $7.2 \times 10^{-5}$ | $1.5 \times 10^{-6}$ |

[a]Average PSD for Winter 8–9 PM. [b]Average PSD for Monsoon 2–3 PM. [c]PSD from Fig. 2 in Laakso et al. (2006). [d]PSD from Table 1 in Hussein et al. (2004).

**Table 3.** Multi-lognormal fitting parameters for the average particle size distribution of each season and characteristic times of day.

| | | Mode 1 | | | | Mode 2 | | | | Mode 3 | | | |
|---|---|---|---|---|---|---|---|---|---|---|---|---|---|
| | | $D_p$ (nm) | $\sigma_g$ | $N_{tot}$ (×10³ cm⁻³) | %PN | $D_p$ (nm) | $\sigma_g$ | $N_{tot}$ (×10³ cm⁻³) | %PN | $D_p$ (nm) | $\sigma_g$ | $N_{tot}$ (×10³ cm⁻³) | %PN |
| 2–3 AM | Winter | 35 | 1.8 | 15.8 | 31 | 122 | 1.8 | 34.5 | 69 | — | — | — | — |
| | Spring | 28 | 1.8 | 20.0 | 43 | 122 | 1.8 | 26.2 | 57 | — | — | — | — |
| | Summer | 23 | 1.6 | 9.1 | 26 | 100 | 1.8 | 26.0 | 74 | — | — | — | — |
| | Monsoon | 26 | 1.8 | 10.2 | 38 | 101 | 1.8 | 16.7 | 62 | — | — | — | — |
| | Autumn | 35 | 1.8 | 10.4 | 28 | 122 | 1.8 | 26.3 | 72 | — | — | — | — |
| 8–9 AM | Winter | 39 | 1.9 | 31.7 | 50 | 122 | 1.9 | 31.8 | 50 | — | — | — | — |
| | Spring | 28 | 1.7 | 37.2 | 61 | 122 | 1.8 | 24.1 | 39 | — | — | — | — |
| | Summer | 21 | 1.6 | 25.4 | 56 | 100 | 1.9 | 20.0 | 44 | — | — | — | — |
| | Monsoon | 23 | 1.6 | 23.5 | 66 | 110 | 1.8 | 12.2 | 34 | — | — | — | — |
| | Autumn | 35 | 1.7 | 26.4 | 58 | 120 | 1.6 | 15.2 | 34 | 234 | 1.6 | 3.6 | 8 |
| 2–3 PM | Winter | 26 | 1.7 | 17.1 | 53 | 125 | 1.8 | 15.0 | 47 | — | — | — | — |
| | Spring | 19 | 1.8 | 41.9 | 81 | 100 | 2.0 | 9.7 | 19 | — | — | — | — |
| | Summer | 26 | 1.9 | 35.5 | 80 | 111 | 1.7 | 8.7 | 20 | — | — | — | — |
| | Monsoon | 27 | 1.7 | 36.5 | 83 | 90 | 1.4 | 4.7 | 11 | 177 | 1.6 | 2.7 | 6 |
| | Autumn | 29 | 1.7 | 14.3 | 56 | 122 | 1.9 | 11.2 | 44 | — | — | — | — |
| 8–9 PM | Winter | 35 | 1.7 | 20.8 | 25 | 88 | 1.9 | 63.0 | 75 | — | — | — | — |
| | Spring | 28 | 1.9 | 38.3 | 55 | 80 | 2.0 | 31.8 | 45 | — | — | — | — |
| | Summer | 30 | 1.8 | 35.9 | 57 | 89 | 1.9 | 27.2 | 43 | — | — | — | — |
| | Monsoon | 27 | 1.6 | 25.4 | 56 | 76 | 1.7 | 17.7 | 39 | 203 | 1.6 | 2.2 | 5 |
| | Autumn | 39 | 1.9 | 24.8 | 48 | 103 | 1.9 | 26.8 | 52 | — | — | — | — |

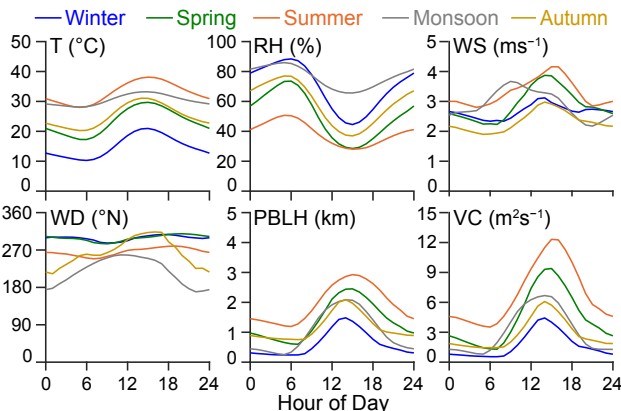

**Figure 1.** Diurnal profiles of meteorological parameters (temperature, relative humidity, wind speed, wind direction, PBLH and VC) by season. Average values by season and hour of day are presented for all parameters except wind direction. The median value is presented for wind direction. Ventilation coefficient (VC) = PBLH × wind speed.

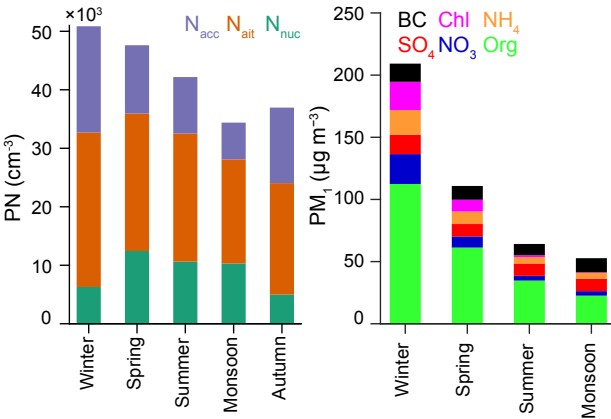

**Figure 2.** Average PN levels for each season by mode. The modes are based on SMPS observations—nucleation (12 < $D_p$ < 25 nm), Aitken (25 < $D_p$ < 100 nm) and accumulation (100 < $D_p$ < 560 nm) modes. The $PM_1$ plot is reproduced from Gani et al. (2019) to illustrate seasonal variation in aerosol mass loadings. The $PM_1$ species are organics (Org), chloride (Chl), ammonium ($NH_4$), nitrate ($NO_3$), sulfate ($SO_4$), and black carbon (BC). While we had SMPS data for all seasons, we did not collect PM composition data during autumn due to instrumentation (ACSM) downtime.

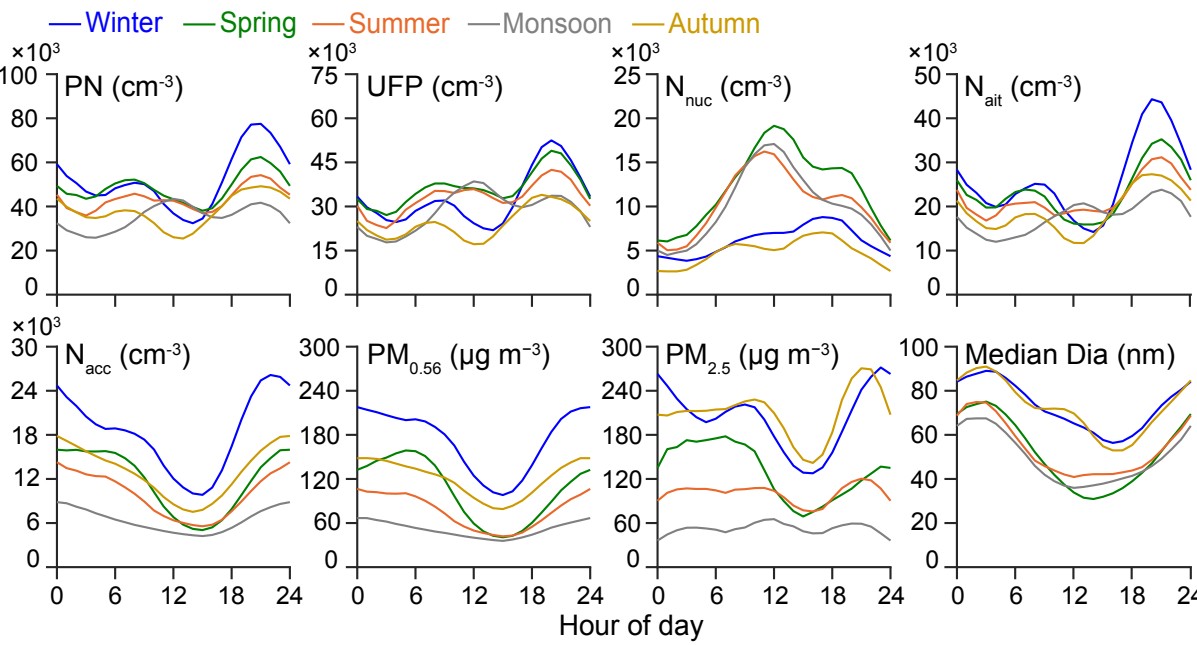

**Figure 3.** Average diurnal and seasonal variations for PN, UFP ($D_p < 100$ nm), $N_{nuc}$ ($D_p < 25$ nm), $N_{ait}$ ($25 < D_p < 100$ nm), $N_{acc}$ ($D_p > 100$ nm), and median particle diameter. These averages are based on the observed SMPS data ($12 < D_p < 560$ nm). Also included are the average diurnal and seasonal variations of the mass concentrations estimated from the observed SMPS data (PM$_{0.56}$) and the PM$_{2.5}$ concentrations from a regulatory monitor (DPCC, R.K. Puram, 3 km from our site).

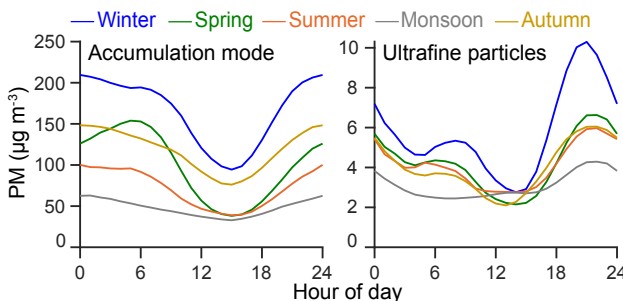

**Figure 4.** Average mass concentrations of observed accumulation mode (100–560 nm) and ultrafine particles (<100 nm) by season and time of day. Estimation of mass concentrations based on assumed particle density of 1.6 g cm$^{-3}$.

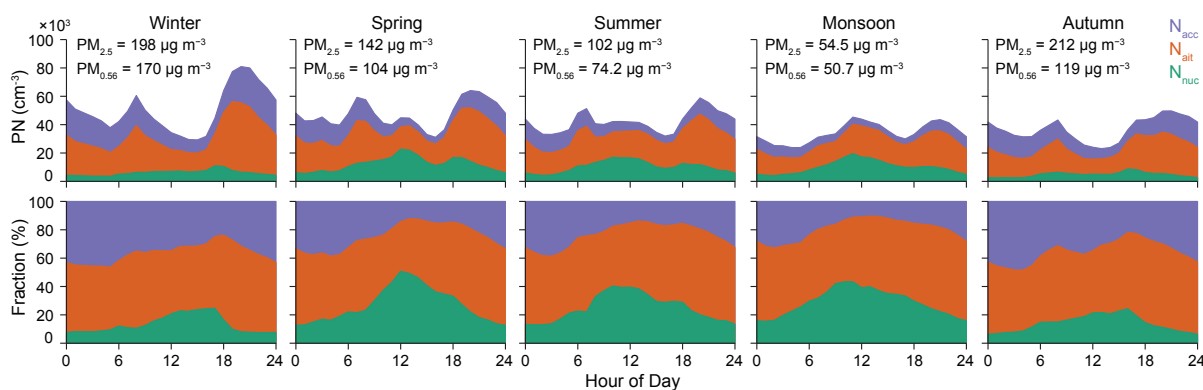

**Figure 5.** Stacked average absolute and fractional diurnal profiles of $N_{nuc}$ ($D_p < 25$ nm), $N_{ait}$ ($25 < D_p < 100$ nm), and $N_{acc}$ ($D_p > 100$ nm) by season. These averages are based on the observed SMPS data ($12 < D_p < 560$ nm). We also include seasonal average concentrations for the $PM_{2.5}$ concentrations from a regulatory monitor (DPCC, R.K. Puram, 3 km from our site) and mass concentrations estimated from the observed SMPS data ($PM_{0.56}$).

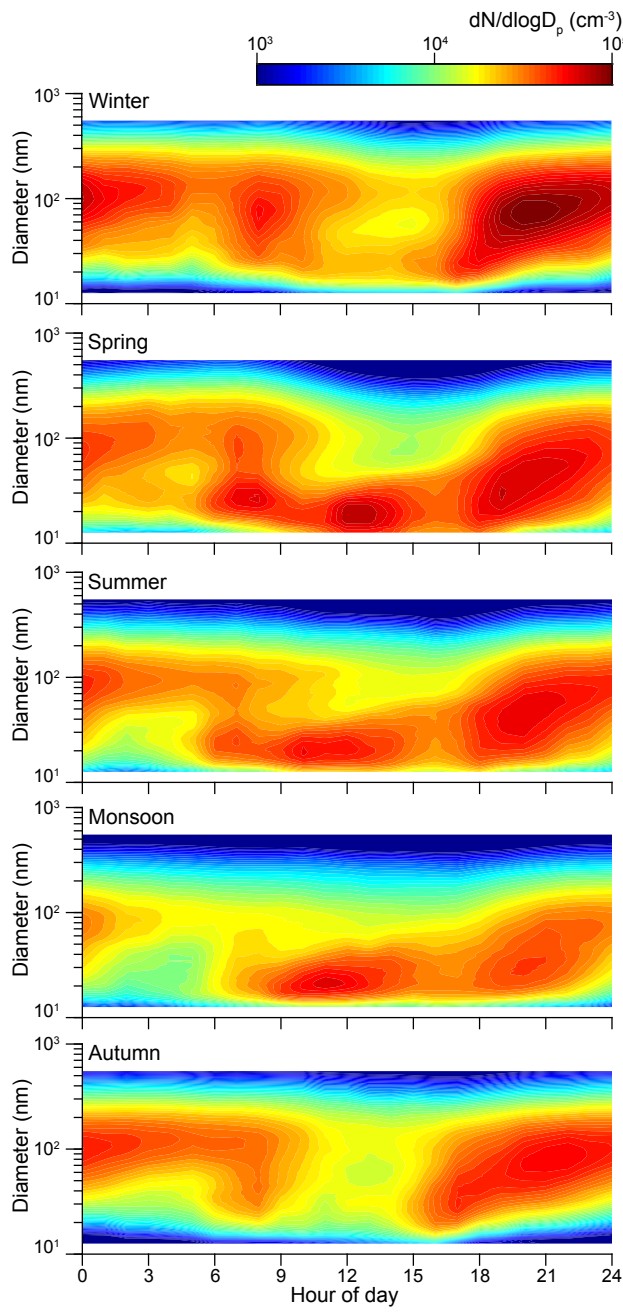

**Figure 6.** Heatmap for particles between 12 and 560 nm averaged for each season.

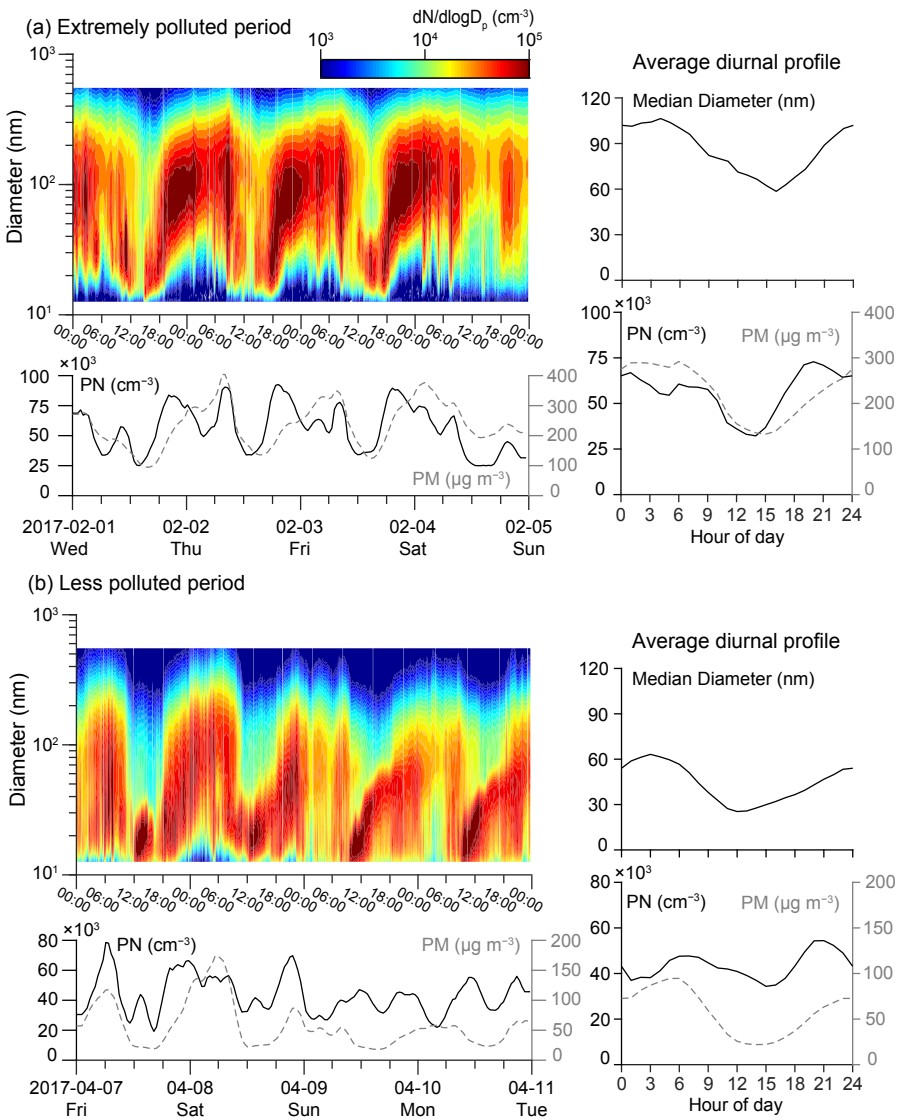

**Figure 7.** Heatmap showing the evolution of the PSD for a (a) polluted period with prominent coagulation scavenging and (b) less polluted period with some new particle formation and growth. Bulk PN and PM concentrations for the same period as the heatmap are also presented for both periods. The average diurnal profile of the median diameter along with the PN and PM concentrations over the two periods are presented in the right panels. Estimation of PM concentrations were based on the volume concentrations observed by the SMPS and assuming particle density to be 1.6 g cm$^{-3}$.

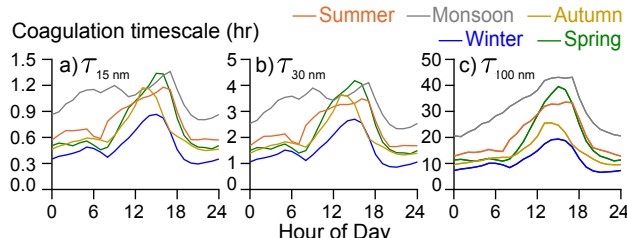

**Figure 8.** Coagulation timescale ($\tau$) for 15, 30 and 100 nm particles by season.