# Peer review of "Particle number concentrations and size distribution in a polluted megacity: The Delhi Aerosol Supersite study"

_Atmospheric Chemistry and Physics, 2020_

## Referee Comment (RC1) · Anonymous Referee #2 · 9 Mar 2020

This manuscript presents a thorough analysis on the behaviour of particle number size distribution, including its connection to the particle mass concentration and associated particle sources and atmospheric processes, in Delhi, India. The topic is important and the paper can be considered original enough to warrant its publication in Atmospheric Chemistry and Physics. The analysis is based on more a year of measurements, allowing the investigation to cover all the relevant seasons. The paper is scientifically sound and well written. There are, however, a few places where interpretation/discussion could be either broadened or improved a bit. My detailed comments in this regard are given below.

The authors explain in detail how they choose the particle density in determining particle mass concentrations from measured particle number size distributions. The total mass concentration is restricted to particles smaller than 560 nm in diameter (PM0.56). Do the authors have any data to tell how well PM0.46 reflects fine particulate matter or PM10 in Delhi? This would be useful information as most mass concentrations discussed in the scientific literature are based on PM2.5 or PM10, or both.

In talking about sub-3 nm particles (page 4, lines 24-28), the authors could include the recent overview by Kontkanen et al (2007, Atmos. Chem. Phys. 17, 2163-87) covering a number of field observations in different types of atmospheric environments.

An interesting feature mentioned at the end of section 3.1 is that ultrafine particle mass concentration in Delhi are comparable to typical PM2.5 mass concentrations in measured in North America. In this context, it would be equally interesting to know how much higher ultrafine particle mass concentrations are in Delhi compared to those in cleaner urban environments in North America, Europe or Asia (for the latter, see. e.g. Bruggemann et al. 2009 Atmos. Environ. 43, 2456-63; Cheung et al. 2016 Atmos. Chem. Phys. 16, 1317-30; Xue et al. 2019 Environ Sci. Technol. 53, 39-49).

When discussing how different factors (insolation, particle loading, RH) influence nucleation mode particles originating from atmospheric new particle formation (lines 12-18 on page 6 and lines 21-22 on page 10), a lot of new information has been obtained since the papers by Kulmala et al (2004) and Hamed et al. (2011), as reviewed in Kerminen et al. (2018, Environ. Res. Lett. 13, 102003). For example, the low particle mass concentrations and high insolation in summer act together to favor nucleation and survival of nucleated particles. Also, while a lower relative humidity seems to favor atmospheric nucleation, the actual reason for this observation has not been fully resolved yet.

The authors mention specific dominating size ranges when discussing particles emitted from different sources (lines 27-32 on page 6, lines 5-7 on page 7 and lines 21-25 on

page 11). Most of the references used in this discussion are 10-20 years old, and may not reflect the current state of knowledge or situation. For examples changes in vehicle engine technology over time or differences the general character of emission sources between India and both Europe and North America probably influences all this. Please check out the discussed "facts" based on more recent literature (e.g. Kumar et al. 2010 Atmos. Environ. 44, 5035-52; Kumar et al. 2013 Atmos. Environ. 67, 252-277; Riffault et al. 2015 Critical Reviews in Environmental Science and Technology 45, 2305-2356; Paasonen et al. 2016 Atmos. Chem. Phys. 16, 6823-40; Zhou et al. 2020 Atmos. Chem. Phys. 20, 1701-1722).

I appreciate the analysis on coagulation scavenging of nucleation mode particles (section 3.3). There are a couple of issues worth to be discussed in more detail here. First, since the lifetime of a nucleation mode particles is sensitive to their diameter, it is essentially a competition between particle growth and scavenging by coagulation that matter the most, not only the lifetime itself (Pierce and Adams 2007 Atmos. Chem. Phys. 7, 1367-79; Lehtinen et al. 2007 J. Aerosol Sci. 38, 988-994). This competition becomes even more important when approaching the sizes at which particles are nucleation, which brings us to the second point: how is it possible that nucleation takes place at all in polluted megacities? This question was investigated by Kulmala et al. (2017, Faraday Discuss. 200, 271-288) but no definite answer could be pointed out. This unsolved issue could be mentioned briefly in this context. Third, there is one more paper related to this topic (Cai and Jiang 2017, Atmos. Chem. Phys. 17, 12659-75) that could be considered here and also when first bringing up the important role of coagulation scavenging ín determining the fate of ultrafine particles (page 2, line 25).

The discussion about the particle growth rates GR (page 9, lines 17-19) need to be revised/updated. First, there are clear differences in typical values (and value ranges) of GR between different types environments, and second, particle growth rates in polluted megacities do not seem to be usually below 5 nm/hour (Nieminen et al. 2018 Atmos. Chem. Phys. 18, 14737-56; Kerminen et al. 2008 Environ. Res. Lett. 13,

102003; Chu et al. 2019 Atmos. Chem. Phys. 19, 115-138).

I do not understand the last statement in section 3.5 (page 10, lines 22-24). Please explain more clearly and justify.

Do the authors have a concrete explanation on why rain would affect much more accumulation mode particles than Aitken or nucleation mode particles (lines 9-11 on page 10)? Is that 1) because accumulation mode particles are scavenged more efficiently by rain compared with smaller particles (is that true at all?), or 2) because accumulation mode particle have overall longer atmospheric lifetimes compared with smaller particles (so that compared with smaller particles, it would take longer to build up an accumulation mode after any individual rain event)?

As a minor technical remark, a correct way to express ranges of quantities is to write "range from M to N", "range between M and N" or "are in the range M-N". Such ranges are incorrectly written in a few places, please correct.

---

## Referee Comment (RC2) · Anonymous Referee #1 · 16 Mar 2020

Delhi is well recognised as one of the world's most polluted megacities and is a city for which there are relatively few data available for highly resolved particle size distributions. This paper reports on around 15 months of continuous measurements of submicrometer particle size distributions and provides an interpretation which gives valuable insights into the processes which determine the size distribution. The length of the dataset also allows segregation according to seasons which is valuable. Calculations of the condensation and coagulation sink are a valuable complement to the data analysis process.

The work is of good quality but there are a number of points which, if addressed, would

significantly enhance the value of the paper. These are as follows:

(a) Sampling took place on the campus of the Indian Institute of Technology in Delhi, but rather little detail is given of the sampling site apart from very local characteristics. The IIT Delhi campus is close to the southern perimeter of Delhi and some comment would be valuable on the degree to which this site could be considered as representative of Delhi more widely. Could we expect to see similar data from a site in the city centre or in Old Delhi? Were the prevailing winds from the city or from the less populated areas to the south?

(b) There are a number of questions concerning the instrumentation. Measurements were made with a TSI Scanning Mobility Particle Sizer. These are generally recommended to be used with an online dryer. Was any conditioning of the inlet air carried out, or do the measurements refer to ambient temperatures and relative humidity? Were the conditions inside the building where the instrument was located the same as those outdoors? Details are given of corrections applied for sampling losses. The TSI instrument software normally calculates sampling losses on the basis of detail of the instrument setup, and if this were carried out, there is a risk of applying two sets of corrections. Was the TSI correction turned off? Which version of the AIM software was used?

(c) The condensation sink was calculated using the properties of sulphur dioxide. Condensation sinks are more often calculated for sulphuric acid vapour and it is not at all clear why SO2 was assumed as the condensing vapour. Table 2 lists condensation and coagulation sinks from the literature referring to studies by Laakso et al. (2006) and Hussein et al. (2004). A search of these papers for the condensing molecule revealed no data for condensation sink or coagulation sink in either paper. This requires further explanation.

(d) The discussion of particle number and mass concentrations and of size-resolved particle concentrations on pages 5-7 does not appear to account for restrictions applying in Delhi for the access of heavy duty vehicles which can only enter certain areas at nighttime. This seems likely to affect the particle number concentrations and size distributions and should be discussed. A further point which is wholly lacking from this part of the discussion is the relative contributions of locally-generated and advected particles. Many workers have highlighted the significance of biomass burning in the rural areas outside of Delhi and there are also highly polluting brick kilns. Can the contributions of such sources be identified? Does the airport (which is quite close) influence the measured data?

(e) On page 7 there is some discussion of comparison of concentrations with other cities but the literature cited is at least 10 years old. Particle number concentrations can change quite massively, as they have done in western Europe as a result of controls on the sulphur content of fuels. Hence, a comparison with such old data is probably of little value and this needs to be updated.

(f) The section on multi-modal PSD fitting draws conclusions based upon assumed patterns of traffic. This would be strengthened by inclusion of a graph showing traffic volumes as a function of time of day, preferably for different vehicle classes.

(g) The comparator used for particle number concentrations in the conclusions section is the review of Kumar et al. (2014). It should be noted that many of the studies included in that review are now very old, and in line with the comment above, it would be more appropriate to compare with more recently collected data.

The manuscript is well written and pleasingly clear of minor typographic errors. However, in Table 1 the units for PM0.56 are incorrect.

---

## Author Comment (AC1) · 25 May 2020

**Reviewer 1:**

**Comments:**

We would like to thank the reviewer for their comments.

Delhi is well recognised as one of the world's most polluted megacities and is a city for which there are relatively few data available for highly resolved particle size distributions. This paper reports on around 15 months of continuous measurements of submicrometer particle size distributions and provides an interpretation which gives valuable insights into the processes which determine the size distribution. The length of the dataset also allows segregation according to seasons which is valuable. Calculations of the condensation and coagulation sink are a valuable complement to the data analysis process.

The work is of good quality but there are a number of points which, if addressed, would significantly enhance the value of the paper. These are as follows:

(a) Sampling took place on the campus of the Indian Institute of Technology in Delhi, but rather little detail is given of the sampling site apart from very local characteristics. The IIT Delhi campus is close to the southern perimeter of Delhi and some comment would be valuable on the degree to which this site could be considered as representative of Delhi more widely. Could we expect to see similar data from a site in the city centre or in Old Delhi? Were the prevailing winds from the city or from the less populated areas to the south?

Thank you for this comment and the opportunity to clarify this important point in the manuscript.

PN concentrations can be extremely variable even within a small spatial domain and definitely in a megacity like Delhi where two different sites can have unique types and strengths of sources emitting in different particle size ranges. While the main themes discussed in the manuscript (eg, rapid coagulation scavenging, suppression of UFP, etc) are most likely site independent given the large concentrations of fine PM across Delhi (and the Indo-Gangetic plain), the absolute PN concentrations can vary among sites in Delhi. Delhi has a prevailing North-western wind and would reach our site in South Delhi after traversing ~20 km of Delhi moving across industrial, commercial, and residential areas.

Added to Section 2.1 (Site details):

"The prevailing wind direction in Delhi is from the northwest, implying that the air we sampled would have traversed through ~20 km of industrial, commercial, and residential areas in Delhi before reaching our site."

Added to Section 2.4 (Limitations and uncertainties):

"Another limitation of this study is that our measurements were limited to a single site, and therefore do not capture the spatial variability within Delhi. UFP concentrations can be quite variable even within a small spatial domain (Saha et al., 2019; Puustinen et al., 2007), so it is

likely that a megacity like Delhi with diverse local sources will have strong spatially variability in PSD and PN concentrations. Future studies can quantify this spatial variability by measuring PSD at multiple fixed sites and using techniques such as mobile monitoring (e.g., Apte et al., 2017)."

(b) There are a number of questions concerning the instrumentation. Measurements were made with a TSI Scanning Mobility Particle Sizer. These are generally recommended to be used with an online dryer. Was any conditioning of the inlet air carried out, or do the measurements refer to ambient temperatures and relative humidity? Were the conditions inside the building where the instrument was located the same as those outdoors? Details are given of corrections applied for sampling losses. The TSI instrument software normally calculates sampling losses on the basis of detail of the instrument setup, and if this were carried out, there is a risk of applying two sets of corrections. Was the TSI correction turned off? Which version of the AIM software was used?

We have added some more information on setup and software to clarify these very relevant questions. Even more details of the setup can be found in Gani et al. (2019).

The sampled air passed through a $PM_{2.5}$ inlet and a Nafion membrane diffusion dryer. The room in which the instruments were installed was air conditioned and generally regulated to ~20–25°C. We used TSI's AIM version 9 software and sampling loss corrections were turned off as we applied our own corrections (discussed in manuscript).

We have included the following text in Sec. 2.2 (Instrumentation and setup) and Sec 2.3 (Data processing and analysis):

"The inlet had a $PM_{2.5}$ cyclone, followed by a water trap and a Nafion membrane diffusion dryer (Magee Scientific sample stream dryer, Berkeley, CA)."

"We used the Aerosol Instrument Manager version 9.0 (TSI, Shoreview, MN) software for logging data from the SMPS (software sampling loss corrections turned off)."

(c) The condensation sink was calculated using the properties of sulphur dioxide. Condensation sinks are more often calculated for sulphuric acid vapour and it is not at all clear why SO2 was assumed as the condensing vapour. Table 2 lists condensation and coagulation sinks from the literature referring to studies by Laakso et al. (2006) and Hussein et al. (2004). A search of these papers for the condensing molecule revealed no data for condensation sink or coagulation sink in either paper. This requires further explanation.

We recalculated the condensation sink for $H_2SO_4$ vapor and updated the values in Table 2 accordingly. Since the updated values calculated for $H_2SO_4$ are only different by ~10% than those calculated earlier for $SO_2$, the discussion remains unchanged.

We used the particle size distribution data from literature to calculate the condensation and coagulation sinks based on these PSDs. The specific PSDs used for these calculations for Delhi and Beijing can be found in Fig. 2 of Laakso et al. (2006) and for Helsinki in Table 1 of Hussein et al. (2004). We have updated the caption and footnotes in Table 2 to reflect these clarifications

and to include the specific figure/table from where we obtained the PSD used for the coagulation and condensation calculations.

Updates to Table 2:

Condensation sink values calculated for $H_2SO_4$ instead of $SO_2$.

Caption: "Characteristic coagulation and condensation sinks for Delhi and comparison with those calculated using PSD in literature from other cities."

Footnotes: "[a]Average PSD for Winter 8–9PM. [b]Average PSD for Monsoon 2–3PM. [c]PSD from Fig.2 in Laakso et al. (2006). [d]PSD from Table 1 in Hussein et al. (2004)."

(d) The discussion of particle number and mass concentrations and of size-resolved particle concentrations on pages 5-7 does not appear to account for restrictions applying in Delhi for the access of heavy duty vehicles which can only enter certain areas at nighttime. This seems likely to affect the particle number concentrations and size distributions and should be discussed. A further point which is wholly lacking from this part of the discussion is the relative contributions of locally-generated and advected particles. Many workers have highlighted the significance of biomass burning in the rural areas outside of Delhi and there are also highly polluting brick kilns. Can the contributions of such sources be identified? Does the airport (which is quite close) influence the measured data?

We have now included some more information on traffic volumes (total and truck traffic) in the manuscript (please refer to response to comment (f)). We agree that biomass burning (both inside and outside Delhi), brick kilns, possibly airport (4km West of our site), and other primary and secondary sources can contribute to number concentrations. While a detailed source apportionment of particle number concentrations is beyond the scope of this manuscript, our previous papers look at source apportionment for aerosol mass (Bhandari et al., 2020; Gani et al., 2019).

(e) On page 7 there is some discussion of comparison of concentrations with other cities but the literature cited is at least 10 years old. Particle number concentrations can change quite massively, as they have done in western Europe as a result of controls on the sulphur content of fuels. Hence, a comparison with such old data is probably of little value and this needs to be updated.

We have updated our literature review to include more recent papers.

"The $N_{ait}$ peaks during the morning and the evening traffic rush hours are consistent with vehicular emissions contributing in this size range (Paasonen et al., 2016; Kumar et al., 2010). Studies from other cities—especially from near-roadway sites—also observe large fraction of the Aitken mode particles in the total PN concentrations (Zhu et al., 2002; Wu et al., 2008). In addition to traffic, other sources contributing to Aitken mode particles include cooking, industrial activities, solid waste burning, and construction activities (Kumar et al., 2013; Riffault et al., 2015; Chen et al., 2017)."

"In addition to emitting in the Aitken mode, biomass burning and older diesel vehicles emit in the accumulation mode as well (Chen et al., 2017; Paasonen et al., 2016; Kumar et al., 2010). Additionally, particles from fresh vehicular emissions can grow rapidly from the Aitken mode at the tailpipe to accumulation mode at roadside and ambient locations (Ning et al., 2013). We speculate that the traffic-related particles in Delhi may get smaller as India's vehicle fleet (especially heavy-duty trucks) is upgraded (Guttikunda and Mohan, 2014; Zhou et al., 2020a)."

(f) The section on multi-modal PSD fitting draws conclusions based upon assumed patterns of traffic. This would be strengthened by inclusion of a graph showing traffic volumes as a function of time of day, preferably for different vehicle classes.

Mishra et al. (2019) reported the traffic conditions at multiple sites in Delhi before and during the period when the odd-even driving scheme (January, 2016). At almost all sites, the morning rush hour was during 8–11am and the evening peak during 5–9pm. Furthermore, there was an increase in heavy duty truck traffic after 10pm (restriction of entry of trucks into Delhi relaxed).

"Delhi traffic usually peaks around 8–11am in the morning and 5–9pm in the evening. Restrictions on daytime heavy duty truck traffic have resulted in increased nocturnal truck traffic after 10pm (Mishra et al., 2019; Guttikunda and Calori, 2013).

(g) The comparator used for particle number concentrations in the conclusions section is the review of Kumar et al. (2014). It should be noted that many of the studies included in that review are now very old, and in line with the comment above, it would be more appropriate to compare with more recently collected data.

We have now generally updated the literature cited throughout the paper. We have removed the 'general PN range' from the conclusions since that would be influenced significantly by the instrumentation size limits (discussed in Sec. 2.4). Instead we have edited/included the following:

"While Delhi experiences PM levels 10 times higher than cities in North America and Western Europe, PN levels were comparable to those observed in urban sites in relatively cleaner (in terms of aerosol mass) cities."

The manuscript is well written and pleasingly clear of minor typographic errors. However, in Table 1 the units for PM0.56 are incorrect.

Typo fixed. Thank you for pointing this out. We have reviewed and updated the manuscript for this and some other typographic errors as well.

References:

[revised manuscript text omitted]

**Reviewer 2:**

We would like to thank the reviewer for their comments.

**Comments:**

This manuscript presents a thorough analysis on the behaviour of particle number size distribution, including its connection to the particle mass concentration and associated particle sources and atmospheric processes, in Delhi, India. The topic is important and the paper can be considered original enough to warrant its publication in Atmospheric Chemistry and Physics. The analysis is based on more a year of measurements, allowing the investigation to cover all the relevant seasons. The paper is scientifically sound and well written. There are, however, a few places where interpretation/discussion could be either broadened or improved a bit. My detailed comments in this regard are given below.

The authors explain in detail how they choose the particle density in determining particle mass concentrations from measured particle number size distributions. The total mass concentration is restricted to particles smaller than 560 nm in diameter (PM0.56). Do the authors have any data to tell how well PM0.46 reflects fine particulate matter or PM10 in Delhi? This would be useful information as most mass concentrations discussed in the scientific literature are based on PM2.5 or PM10, or both.

We have now included a comparison of SMPS-$PM_{0.56}$ and $PM_{2.5}$ to the supplement (Fig. S14). Slope = 0.60 and $R^2$ = 0.60 for the least square fit between SMPS-$PM_{0.56}$ and $PM_{2.5}$. The $PM_{2.5}$ data used for this comparison was based on a regulatory site 3 km from our site. We have also included a comparison of the C-$PM_1$ (Composition-$PM_1$ = NR-$PM_1$ + BC) and SMPS-$PM_1$ in Fig. S14 (Slope = 0.95 and $R^2$ = 0.83). SMPS-$PM_1$ was calculated by extrapolating the PSD to 1μm using PSD model fitting and C-$PM_1$ was based on data from an ACSM and Aethalometer at our site (Gani et al., 2019).

In talking about sub-3 nm particles (page 4, lines 24-28), the authors could include the recent overview by Kontkanen et al (2007, Atmos. Chem. Phys. 17, 2163-87) covering a number of field observations in different types of atmospheric environments.

Included citation for Kontkanen et al. (2017) as it is indeed pertinent to the discussion.

An interesting feature mentioned at the end of section 3.1 is that ultrafine particle mass concentration in Delhi are comparable to typical PM2.5 mass concentrations in measured in North America. In this context, it would be equally interesting to know how much higher ultrafine particle mass concentrations are in Delhi compared to those in cleaner urban environments in North America, Europe or Asia (for the latter, see. e.g. Bruggemann et al. 2009 Atmos. Environ. 43, 2456-63; Cheung et al. 2016 Atmos. Chem. Phys. 16, 1317-30; Xue et al. 2019 Environ Sci. Technol. 53, 39-49).

We have now included these studies with the following text:

"Overall the mass contributions of UFP ranged between 2.1 and 10.3 µg m$^{-3}$ depending on season and time of day. For contrast, mass concentrations of UFP have been observed to be <3 µg m$^{-3}$ in Germany (Brüggemann et al., 2009), <2 µg m$^{-3}$ in Taiwan (Cheung et al., 2016), and <0.2 µg m$^{-3}$ in California (Xue et al., 2019)."

When discussing how different factors (insolation, particle loading, RH) influence nucleation mode particles originating from atmospheric new particle formation (lines 12-18 on page 6 and lines 21-22 on page 10), a lot of new information has been obtained since the papers by Kulmala et al (2004) and Hamed et al. (2011), as reviewed in Kerminen et al. (2018, Environ. Res. Lett. 13, 102003). For example, the low particle mass concentrations and high insolation in summer act together to favor nucleation and survival of nucleated particles. Also, while a lower relative humidity seems to favor atmospheric nucleation, the actual reason for this observation has not been fully resolved yet.

We have now included this recent review paper along with following updates to the text in both sections mentioned:

 "The low particle mass concentrations and high insolation during the daytime of warmer months act together to favor nucleation and survival of nucleated particles (Kerminen et al., 2018), resulting in higher N$_{nuc}$ concentrations during these periods."

"In addition to large condensation sinks from existing particles (Sec. 3.4), it is possible that the elevated RH during the cooler winter and autumn (Fig. 1) could further suppress atmospheric nucleation. It has been observed in both clean and polluted environments that the RH during new particle formation event days tends to be lower compared to non-event days (Kerminen et al., 2018; Dada et al., 2017; Hamed et al., 2011).

The authors mention specific dominating size ranges when discussing particles emitted from different sources (lines 27-32 on page 6, lines 5-7 on page 7 and lines 21-25 on page 11). Most of the references used in this discussion are 10-20 years old, and may not reflect the current state of knowledge or situation. For examples changes in vehicle engine technology over time or differences the general character of emission sources between India and both Europe and North America probably influences all this. Please check out the discussed "facts" based on more recent literature (e.g. Kumar et al. 2010 Atmos. Environ. 44, 5035-52; Kumar et al. 2013 Atmos. Environ. 67, 252-277; Riffault et al. 2015 Critical Reviews in Environmental Science and Technology 45, 2305-2356; Paasonen et al. 2016 Atmos. Chem. Phys. 16, 6823-40; Zhou et al. 2020 Atmos. Chem. Phys. 20, 1701-1722).

We have updated the references to incorporate suggestions from both reviewers and a few more including those specifically suggested in this comment (Kumar et al., 2010; Kumar et al., 2013; Riffault et al., 2015; Paasonen et al., 2016; Zhou et al., 2020).

The edits to the manuscript in the order mentioned in the comment by the reviewer are as follows:

"The $N_{ait}$ peaks during the morning and the evening traffic rush hours are consistent with vehicular emissions contributing in this size range (Paasonen et al., 2016; Kumar et al., 2010). Studies from other cities—especially from near-roadway sites—also observe large fraction of the Aitken mode particles in the total PN concentrations (Zhu et al., 2002; Wu et al., 2008). In addition to traffic, other sources contributing to Aitken mode particles include cooking, industrial activities, solid waste burning, and construction activities (Kumar et al., 2013; Riffault et al., 2015; Chen et al., 2017)."

"In addition to emitting in the Aitken mode, biomass burning and older diesel vehicles emit in the accumulation mode as well (Chen et al., 2017; Paasonen et al., 2016; Kumar et al., 2010). Additionally, particles from fresh vehicular emissions can grow rapidly from the Aitken mode at the tailpipe to accumulation mode at roadside and ambient locations (Ning et al., 2013). We speculate that the traffic-related particles in Delhi may get smaller as India's vehicle fleet (especially heavy-duty trucks) is upgraded (Guttikunda and Mohan, 2014; Zhou et al., 2020a)."

"The relatively smaller median diameter for the accumulation mode during the traffic rush hour is consistent with particles from fresh vehicle exhaust—both gasoline and diesel—being smaller than those from biomass burning and other aged aerosol. While gasoline engines emit in the ~20–60 nm range, some of the older diesel engines still prevalent in India can emit larger accumulation mode particles (Kumar et al., 2010). Particles from biomass burning are in the ~50–200 nm size range (Chen et al., 2017; Janhäll et al., 2010). For the warmer periods which have relatively less biomass burning emissions, the ultrafine modes contribute to relatively higher fraction of PN—especially during traffic rush hours."

I appreciate the analysis on coagulation scavenging of nucleation mode particles (section 3.3). There are a couple of issues worth to be discussed in more detail here. First, since the lifetime of a nucleation mode particles is sensitive to their diameter, it is essentially a competition between particle growth and scavenging by coagulation that matter the most, not only the lifetime itself (Pierce and Adams 2007 Atmos. Chem. Phys. 7, 1367-79; Lehtinen et al. 2007 J. Aerosol Sci. 38, 988-994). This competition becomes even more important when approaching the sizes at which particles are nucleation, which brings us to the second point: how is it possible that nucleation takes place at all in polluted megacities? This question was investigated by Kulmala et al. (2017, Faraday Discuss. 200, 271-288) but no definite answer could be pointed out. This unsolved issue could be mentioned briefly in this context. Third, there is one more paper related to this topic (Cai and Jiang 2017, Atmos. Chem. Phys. 17, 12659-75) that could be considered here and also when first bringing up the important role of coagulation scavenging in determining the fate of ultrafine particles (page 2, line 25).

We agree with the reviewer about the competition between growth rate and the coagulation scavenging. We calculated the survival parameter defined in Kulmala et al. (2017) and found that even for the least polluted period in Delhi we should not be able to observe NPF (P>50 for the least polluted period). This finding was consistent with the discussion on megacities in China in Kulmala et al. (2017). We have included the following text in our discussion on NPF (end of Section 3.4):

"Based on the survival parameter ($P = (CS/10^4 \text{ s}^{-1})/(GR/\text{nm h}^{-1})$) defined in Kulmala et al. (2017), new particle formation would not be expected even during some of the least polluted periods in Delhi. Why we observe any new particle formation events in Delhi—similar to observations from megacities in China—is still not well understood (Kulmala et al., 2017)."

We have also added Cai and Jiang (2017) to the discussion on coagulation scavenging in determining the fate of UFP in China (Section 1).

The discussion about the particle growth rates GR (page 9, lines 17-19) need to be revised/updated. First, there are clear differences in typical values (and value ranges) of GR between different types environments, and second, particle growth rates in polluted megacities do not seem to be usually below 5 nm/hour (Nieminen et al. 2018 Atmos. Chem. Phys. 18, 14737-56; Kerminen et al. 2008 Environ. Res. Lett. 13, 102003; Chu et al. 2019 Atmos. Chem. Phys. 19, 115-138).

We agree with the reviewer. We have revised this discussion based on latest literature. We have also rephrased our discussion around the absence of new particle formation and growth during the polluted periods observed in Delhi.

"The growth rate of the nucleation mode particles based on the average PSD for these clean conditions was ~5 nm h$^{-1}$. It should be noted that our SMPS measurements had a lower size cut-off of 12 nm, implying the particles in the nucleation mode had already grown before being detected.

Growth rates depend on particle size, concentrations of condensable gases (H$_2$SO$_4$, highly oxygenated molecules, etc), and condensation and coagulation sinks, and they can range from 0.1 to 10s of nm h$^{-1}$ across various environments (Kerminen et al., 2018; Nieminen et al., 2018; Bianchi et al., 2016). We did not observe banana-shaped growth patterns during the polluted winter or autumn seasons as the growth phase was probably disrupted by coagulation scavenging. For example, during winter, coagulation timescales for a 15 nm particle were <1 h (Fig. 8). Growth rates for polluted megacities also usually slower than 10 nm h−1 (Zhou et al., 2020b; Chu et al., 2019). It is likely that a nucleation mode particle is more likely to coagulate with a larger particle than to grow to the Aitken/accumulation mode. Overall, the aerosol dynamics are a complex interplay of both new particle formation and coagulation scavenging, with nucleation mode particles in highly polluted environments generally susceptible to scavenging by the accumulation mode particles."

I do not understand the last statement in section 3.5 (page 10, lines 22-24). Please explain more clearly and justify.

We have now added some more context to this concluding statement of that section.

Meteorology, specifically changes in the ventilation coefficient, have a strong influence in the seasonal difference in aerosol mass loadings in Delhi. However, we do not observe this strong influence of meteorology on particle number concentrations. While warmer periods (eg, summer) have lower mass loadings, the lower condensation/coagulation sink along with the

higher insolation results in higher number concentrations. The cooler polluted periods (eg, winter) have higher aerosol mass loadings and consequently a larger condensation/coagulation sink, suppressing the number concentrations. So, while ventilation coefficient has an inverse relationship with the mass loadings (ventilation higher = lower PM concentrations), ventilation coefficient does not influence number concentrations in the same straightforward manner that it modulates mass concentrations.

The new concluding statements for Section 3.5:

"Overall, ventilation strongly influences aerosol mass loadings in Delhi, with higher ventilation often resulting in reduction of PM concentrations and vice versa. However, we do not observe such a simple relationship between ventilation and PN levels. The complex interplay of sources, aerosol dynamics, and atmospheric processes as discussed above make PN levels less sensitive to meteorological changes as compared to aerosol mass loadings."

Do the authors have a concrete explanation on why rain would affect much more accumulation mode particles than Aitken or nucleation mode particles (lines 9-11 on page 10)? Is that 1) because accumulation mode particles are scavenged more efficiently by rain compared with smaller particles (is that true at all?), or 2) because accumulation mode particle have overall longer atmospheric lifetimes compared with smaller particles (so that compared with smaller particles, it would take longer to build up an accumulation mode after any individual rain event)?

Based on our observations, there does not seem to be a clear reduction in UFP concentrations during the rain event (at least not to the degree observed for accumulation mode particles). The lower reduction of the Aitken mode particles seems to be consistent with parametrization studies (Pryor et al., 2016; Laakso et al., 2003). However, these studies suggest that the nucleation mode particles should be scavenged more than Aitken mode particles. It is possible that the rain washout of nucleation mode particles is compensated by the reduction in coagulation/condensation sink resulting from a reduction in the accumulation mode particles.

"Comparing rainy and non-rainy days during the monsoon, our observations suggest that rain reduces the accumulation mode concentrations (Fig. S12) as has been observed in other studies (Luan et al., 2019; Hussein et al., 2018). However, we do not observe a similar reduction in either nucleation or Aitken mode particle concentrations. Parametrization studies suggest that rain is more effective in scavenging nucleation mode particles as compared to Aitken mode particles (Pryor et al., 2016; Laakso et al., 2003). While we cannot fully reconcile our observations here with theory, possible explanations include (i) differential water solubility by particle size and age, or (ii) that the rain washout of the nucleation mode particles is 'balanced' by the reduction in the condensation and coagulation sink due to rain washout of accumulation mode particles."

As a minor technical remark, a correct way to express ranges of quantities is to write "range from M to N", "range between M and N" or "are in the range M-N". Such ranges are incorrectly written in a few places, please correct.

Thank you for bringing this point to our attention. We have updated the manuscript to make it consistent with this remark.

References:

[revised manuscript text omitted]